# Discrimination of Bacterial Community Structures among Healthy, Gingivitis, and Periodontitis Statuses through Integrated Metatranscriptomic and Network Analyses

Takashi Nemoto,[a] Takahiko Shiba,[a] Keiji Komatsu,[a] Takayasu Watanabe,[b] Masahiro Shimogishi,[c] Masaki Shibasaki,[c] Tatsuro Koyanagi,[a] Takahiko Nagai,[a] Sayaka Katagiri,[a] Yasuo Takeuchi,[a] Takanori Iwata[a]

[a]Department of Periodontology, Graduate School of Medical and Dental Sciences, Tokyo Medical and Dental University (TMDU), Tokyo, Japan
[b]Department of Chemistry, Nihon University School of Dentistry, Tokyo, Japan
[c]Department of Regenerative and Reconstructive Dental Medicine, Graduate School of Medical and Dental Sciences, Tokyo Medical and Dental University (TMDU), Tokyo, Japan

**ABSTRACT** Periodontal disease is an inflammatory condition caused by polymicrobial infection. The inflammation is initiated at the gingiva (gingivitis) and then extends to the alveolar bone, leading to tooth loss (periodontitis). Previous studies have shown differences in bacterial composition between periodontal healthy and diseased sites. However, bacterial metabolic activities during the health-to-periodontitis microbiome shift are still inadequately understood. This study was performed to investigate the bacterial characteristics of healthy, gingivitis, and periodontitis statuses through metatranscriptomic analysis. Subgingival plaque samples of healthy, gingivitis, and periodontitis sites in the same oral cavity were collected from 21 patients. Bacterial compositions were then determined based on 16S rRNA reads; taxonomic and functional profiles derived from genes based on mRNA reads were estimated. The results showed clear differences in bacterial compositions and functional profiles between healthy and periodontitis sites. Co-occurrence networks were constructed for each group by connecting two bacterial species if their mRNA abundances were positively correlated. The clustering coefficient values were 0.536 for healthy, 0.600 for gingivitis, and 0.371 for periodontitis sites; thus, network complexity increased during gingivitis development, whereas it decreased during progression to periodontitis. Taxa, including *Eubacterium nodatum*, *Eubacterium saphenum*, *Filifactor alocis*, and *Fretibacterium fastidiosum*, showed greater transcriptional activities than those of red complex bacteria, in conjunction with disease progression. These taxa were associated with periodontal disease progression, and the health-to-periodontitis microbiome shift was accompanied by alterations in bacterial network structure and complexity.

**IMPORTANCE** The characteristics of the periodontal microbiome influence clinical periodontal status. Gingivitis involves reversible gingival inflammation without alveolar bone resorption. In contrast, periodontitis is an irreversible disease characterized by inflammatory destruction in both soft and hard tissues. An imbalance of the microbiome is present in both gingivitis and periodontitis. However, differences in microbiomes and their functional activities in the healthy, gingivitis, and periodontitis statuses are still inadequately understood. Furthermore, some inflamed gingival statuses do not consistently cause attachment loss. In this study, metatranscriptomic analyses were used to investigate the specific bacterial composition and gene expression patterns of the microbiomes of the healthy, gingivitis, and periodontitis statuses. In addition, co-occurrence network analysis revealed that the gingivitis site included features of networks observed in both the healthy and periodontitis sites. These results provide transcriptomic evidence to support gingivitis as an intermediate state between the healthy and periodontitis statuses.

Address correspondence to Takahiko Shiba, shiba.peri@tmd.ac.jp, or Yasuo Takeuchi, takeuchi.peri@tmd.ac.jp.

**KEYWORDS** gingivitis, periodontitis, metatranscriptome, co-occurrence network, oral microbiome, dysbiosis

The periodontium comprises the soft tissue and bone surrounding the tooth, and periodontal diseases are representative polymicrobial diseases that involve a microbiome imbalance known as dysbiosis, which triggers periodontal inflammation (1, 2). Gingivitis is a reversible disease that comprises local gingival inflammation without the loss of connective tissue attachment or alveolar bone (3), while periodontitis is an irreversible disease characterized by inflammatory destruction in both soft and hard tissues (4, 5). Periodontitis leads to tooth loss and oral functional decline, and there is increasing evidence that it is associated with the onset or progression of various systemic diseases (5–9).

As a major etiologic factor in periodontal diseases, the presence of specific bacteria has been extensively investigated to characterize links with periodontal disease progression. Numerous studies have shown that specific Gram-negative anaerobic bacteria and/or their toxic products are associated with inflammatory destruction in periodontal disease (10–12); however, most of these oral bacteria have not yet been cultivated. The availability of new and improved sequencing technologies has expanded the understanding of the periodontal microbial ecosystem (13), Notably, some studies using DNA-based target sequencing and whole-genome sequencing assays have revealed differences in species composition and function between periodontally healthy and disease-associated microbiomes (4, 14). However, these analyses are unable to differentiate between live and dead cells, thus potentially overestimating the bacterial community richness and leading to misunderstandings concerning bacterial characteristics (15). RNA-based sequencing analyses can identify only live and metabolically active cells (16–18). Some studies have shown differences in bacterial composition and gene expression between periodontally healthy and diseased sites by using metatranscriptomic sequencing (19, 20). Jorth et al. compared microbiomes between periodontitis and healthy sites; they found that *Fusobacterium nucleatum* was associated with butyrate metabolism, which led to a pathological periodontal environment (19). Nowicki et al. showed the increased abundance of some genera (e.g., *Oribacterium* and *Leptotrichia*) and the overexpression of virulence-related genes in *Leptotrichia buccalis*, *Prevotella nigrescens*, and *F. nucleatum* in gingivitis (20). However, there remains an inadequate understanding regarding the specific bacterial composition and/or gene expression patterns of the microbiomes during the alteration from health to periodontitis, because microbiome variations related to individual differences were not considered in most previous studies (21). In this study, we compared microbiomes among the healthy, gingivitis, and periodontitis statuses to clarify bacterial features involved in the onset and progression of periodontal diseases. To our knowledge, this is the first study to use metatranscriptomic analysis to investigate differences in bacterial composition and bacterial gene expression profiles among the healthy, gingivitis, and periodontitis statuses within a single oral cavity.

## RESULTS

**Clinical characteristics of participants.** In total, 21 patients (7 men [including 1 smoker] and 14 women) were recruited for this study. The mean age was 61.2 years (range, 36 to 88 years). The clinical characteristics of the participants are summarized in Table 1. Both mean probing depth (PD) and radiographic bone loss were significantly greater at periodontitis sites than at healthy or gingivitis sites (referred to as the P, H, and G sites throughout Results).

**Evaluation of bacterial compositions based on 16S rRNA sequences.** In total, 43,487,892 sequence reads were generated, corresponding to a mean of 690,284 (range, 71,876 to 1,590,068) reads per sample. The mean numbers of reconstructed 16S rRNA (here referred to as rc-rRNA) operational taxonomic units (OTUs) were 37.3 ± 18.6, 45.5 ± 21.8, and 37.6 ± 26.2 in the H, G, and P sites, respectively (see

**TABLE 1** Clinical characteristics of study participants[a]

| Site | PD (mm) | Radiographic bone loss (mm) | BOP (% of patients) |
|---|---|---|---|
| Healthy | 2.4 ± 0.6† | 2.70 ± 1.08† | 0 |
| Gingivitis | 2.6 ± 0.5* | 2.76 ± 1.08* | 100 |
| Periodontitis | 6.4 ± 1.3*† | 5.98 ± 1.78*† | 100 |

[a]Values represent means ± standard deviations. The mean age of patients was 61.2 ± 15.3 years. Seven patients were male, and 14 were female. PD, probing depth; BOP, bleeding on probing. *, significant difference between gingivitis and periodontitis sites; †, significant difference between healthy and periodontitis sites.

Fig. S1A in the supplemental material). There were no significant differences in OTU numbers ($P = 0.165$) and Shannon indexes ($P = 0.156$), which were used to compare alpha diversities among the three periodontal statuses (Fig. S1A and B). Rarefaction curve assessment indicated that the number of obtained reads was sufficient for 16S rRNA analyses (Fig. S1C).

The rc-rRNAs were assigned to 77 genera; 62, 70, and 62 genera were identified in the H, G, and P sites, respectively. Analysis of bacterial composition at the genus level showed that *Porphyromonas* was the predominant genus in the P site (Fig. S2). In contrast, the predominant genera in the H and G sites varied among patients. At the species level, the rc-rRNAs were assigned to 225 bacterial taxa; 168, 192, and 157 taxa were identified in the H, G, and P sites, respectively (Table S1A). The total abundance of the red complex species (*Porphyromonas gingivalis*, *Tannerella forsythia*, and *Treponema denticola*) (10) was greater in the P (34%) site than in the H (4%) and G (13%) sites (Fig. 1A). Relative 16S rRNA abundances of 35 species that comprised >50% of samples significantly differed among the three periodontal statuses; differences were observed mainly in comparisons of the P site with the H or G site. The abundances of *Desulfobulbus* sp., *Eubacterium* [XI][G-5] *saphenum*, *Filifactor alocis*, *Fretibacterium fastidiosum*, *Fretibacterium* sp., *Mogibacterium timidum*, *P. gingivalis*, *T. forsythia*, *T. denticola*, and *Treponema* sp. were significantly greater in P sites than in H or G sites.

Principal-coordinate analysis (PCoA) plots based on Bray-Curtis dissimilarity were constructed to examine differences in beta diversity among the three periodontal statuses (Fig. 1B). These results were supported by permutational multivariate analysis of variance (PERMANOVA), which revealed that bacterial compositions were dissimilar among the three periodontal statuses ($F = 5.86$ and $P = 1.00E–4$). PCoA confirmed that all PCoA plots from the smoker were within the standard deviation zone for each of the three periodontal statuses.

**Comparison of functional profiles among microbiomes.** Using the SEED subsystems, 2,481, 2,636, and 2,050 bacterial genes were assigned to the H, G, and P sites, respectively (Table S1B). The numbers of bacterial genes did not significantly differ among the three periodontal statuses ($P = 0.101$); 1,575 bacterial genes were commonly expressed among all three periodontal statuses. Among the level 1 SEED subsystems, "Protein Metabolism" (H, 26.5%; G, 26.0%; and P, 19.3%), "RNA Metabolism" (H, 9.1%; G, 12.9%; and P, 28.1%), "Clustering-based subsystems" (H, 13.0%; G, 12.1%; and P, 9.2%), and "Carbohydrates" (H, 12.8%; G, 11.9%; and P, 7.8%) were predominant in all three periodontal statuses (Fig. 2A). PCoA (Fig. 2B) and PERMANOVA confirmed the significant difference in the level 1 SEED subsystem composition among the three periodontal statuses ($F = 17.06$ and $P = 1.00E–4$). Sixteen functional categories of the level 1 SEED subsystems significantly differed among the three periodontal statuses. Some functions (e.g., branched-chain amino acids, fermentation, flagellar motility, and amino acid metabolism) showed significantly greater expression in P sites than in H or G sites.

Based on analysis using the Kyoto Encyclopedia of Genes and Genomes (KEGG) database, 1,489, 1,447, and 1,079 bacterial genes were assigned to the H, G, and P sites, respectively (Table S1C). The numbers of bacterial genes did not significantly differ among the three periodontal statuses ($P = 0.097$). In total, 883 bacterial genes were commonly expressed among all three periodontal statuses; 310, 232, and 72 genes

**(A)**

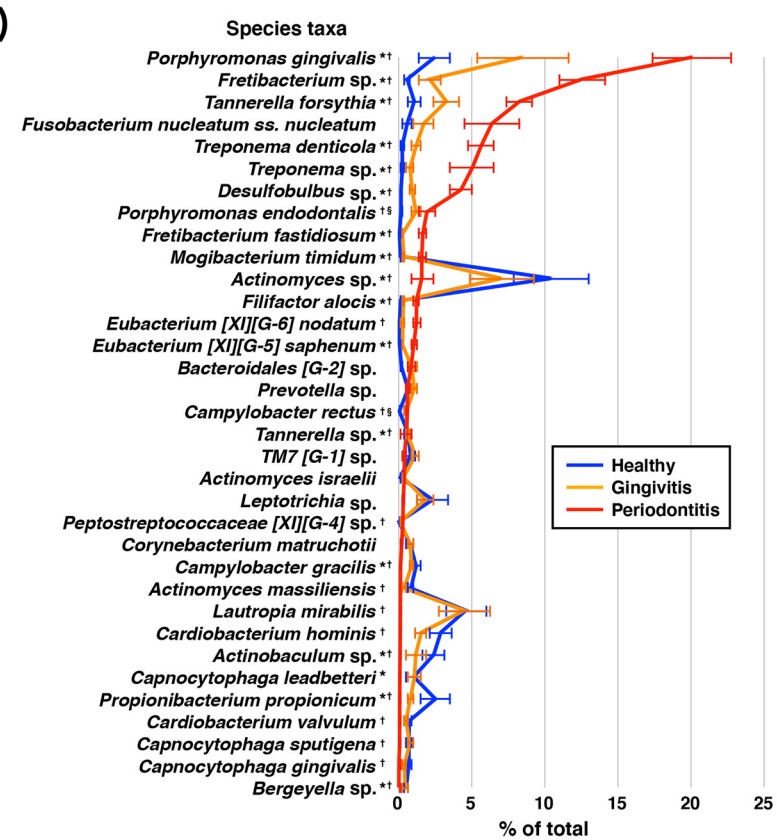

**(B)**

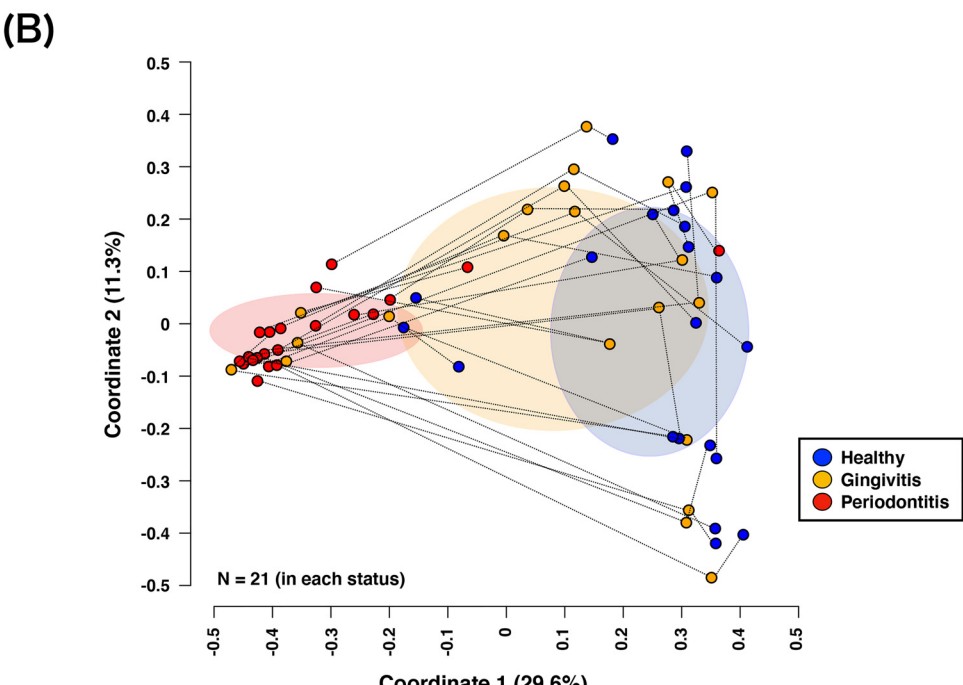

**FIG 1** Relative 16S rRNA abundances of bacterial taxa and PCoA plot of 16S rRNA profiles. (A) Mean rank distributions of the taxonomic origins of reconstructed 16S rRNA (rc-rRNA) clusters in healthy, gingivitis, and periodontitis sites. Mean rc-rRNA abundances detected in >11 samples (>50% of samples) for any status are shown in descending order. The bars show mean ± standard error (SE) relative abundances. *, significant difference between gingivitis and periodontitis site; †, significant difference between healthy and periodontitis sites; §, significant difference between healthy and gingivitis sites. (B) PCoA was conducted to examine the Bray-Curtis dissimilarity; 21 samples from healthy, gingivitis, and periodontitis sites were plotted

were identified solely in H, G, and P sites, respectively. Moreover, 44, 252, and 80 genes were identified only in G and P sites, only in H and G sites, and only in H and P sites, respectively. PCoA (Fig. 2C) and PERMANOVA revealed dissimilar functional profiles among the three periodontal statuses ($F = 2.21$ and $P = 0.045$). Some functions (e.g., fatty acid biosynthesis, lipopolysaccharide biosynthesis, carbon fixation pathways, bacterial chemotaxis, and flagellar assembly) had significantly greater expression in P sites than in H or G sites. Functional pathways detected in 11 or more samples were visualized (Fig. 2D) to reveal the predominant pathways that were active in H, G, and P sites; 37 pathways (e.g., tricarboxylic acid cycle, fatty acid biosynthesis and degradation, methane metabolism, and amino acid metabolism) were predominant in all three periodontal statuses. In contrast, five pathways (benzoate degradation, dioxin degradation, synthesis and degradation of ketone bodies, tryptophan metabolism, and xylene degradation) were active only in G and P sites. Five other pathways (D-alanine metabolism, D-glutamine and D-glutamate metabolism, inositol phosphate metabolism, nitrogen metabolism, and selenocompound metabolism) were active only in H and G sites. Functional profiles estimated based on the National Center for Biotechnology Information nonredundant (NCBI nr) protein database, the Virulence Factors of Pathogenic Bacteria database, and MvirDB are described in Text S1.

**Characterization of taxonomic mRNA origins and detection of viable taxa with high mRNA abundance.** A previous metatranscriptomic study of the oral cavity showed that the 16S rRNA sequence-based bacterial composition differed from the taxonomic mRNA profile-based bacterial composition (17, 18). Using data that were functionally annotated from the NCBI nr protein database, we first assessed the taxonomic origin of each gene identified in the mRNA clusters (Fig. S3 and Table S1D). The total numbers of taxa at the species level in H, G, and P sites were 2,058, 2,279, and 2,129, respectively; in total, 1,228 taxa were commonly detected among all three periodontal statuses. In P sites, the most predominant taxon was *P. gingivalis*; conversely, *Actinomyces* sp. was the most predominant taxon in H and G sites. Assessments using PCoA and analysis of similarities (ANOSIM) showed differences in the bacterial compositions of mRNA and rc-rRNA clusters based on read abundances in each site (H site, $R = 0.130$, $P = 1.00E–4$; G site, $R = 0.135$, $P = 6.00E–4$; and P site, $R = 0.267$, $P = 1.00E–4$) (Fig. S4). Taxa detected in both the rc-rRNA and mRNA profiles were defined as viable taxa with *in situ* functions (VTiF), and VTiF with an mRNA/16S rRNA read abundance of $>1$ were used to define active taxa with high activity. In total, 122, 139, and 112 VTiF were identified in the H, G, and P sites, respectively. Of these, 116, 134, and 105 VTiF in the H, G, and P sites, respectively, were regarded as active taxa. Among active taxa, the activities of four taxa (*F. fastidiosum*, *Eubacterium nodatum*, *F. alocis*, and *Prevotella* sp.) were greater than the activities of red complex bacteria in P sites. We also defined "significant" active taxa as taxa in which the read abundance of mRNA was significantly higher than the read abundance of rc-rRNA. The results showed 26, 45, and 20 significant active taxa in the H, G, and P sites, respectively (Fig. 3A and Table S2). Rates of enhancement of mRNA/16S rRNA ratios for each taxon are shown in Fig. 3B. Comparison of mRNA/16S rRNA ratios between the H and G sites revealed that *F. fastidiosum* showed the greatest rate of enhancement; *E. nodatum* showed the greatest rate of enhancement in the comparison between G and P sites.

**VTiF in co-occurrence networks and interacting core taxa.** Co-occurrence network analysis using the SparCC correlation coefficient showed 36, 45, and 26 nodes, as well as 3.06, 3.15, and 1.73 mean edges per node, in the H, G, and P sites, respectively (Fig. 4A). Clustering coefficients were 0.536, 0.600, and 0.371 in the H, G, and P sites, respectively. The significant active taxa (26, 45, and 20 taxa in the H, G, and P sites, respectively) were prevalent in each network (Fig. 4A, Table 2, and Table S3). Some sig-

**FIG 1** Legend (Continued)
with three coordinates. The mean and standard deviation in each axis are indicated by an ellipse for each status. Dots corresponding to three periodontal statuses from the same patient are connected by a broken line.

**FIG 2** mRNA profiles obtained following assignments with the SEED subsystems and the KEGG database. (A) Mean rank distributions of functional categories based on the level 1 SEED subsystems in healthy, gingivitis, and periodontitis sites. The bars show mean ±

nificant active taxa also exhibited significant positive SparCC correlations; these taxa were regarded as interacting core taxa. Although two interacting core taxa were present in P sites, more interacting core taxa were detected in H sites (nine interacting core taxa) and G sites (23 interacting core taxa) (Fig. 4A and B). In addition, the gingivitis co-occurrence network appeared to comprise three subnetworks that included features of healthy and periodontitis microbiomes (Fig. 4A). Subgroup 1 included mainly taxa that exhibited co-occurrence relationships with red complex bacteria; these taxa were detected in both G and P sites. The subgroup 2 subnetwork included mainly taxa that were detected only in G sites. Subgroup 3 included mainly taxa that were detected in both H and G sites.

## DISCUSSION

Periodontitis is a highly prevalent infectious disease among adults, especially seniors, worldwide. Approximately 10.8% of the global population (743 million people) has been affected by severe periodontitis (22). Although the presence of highly pathogenic bacteria and virulence factors has been reported, changes in biofilm composition alone do not explain the onset or development of periodontal diseases. A better understanding of the alteration of the bacterial composition together with the metabolic activities of the microbiome during early and progressed states of periodontal disease will help to identify the underlying cause of this disease (21); this understanding may also help to establish a targeted approach to suppress disease progression. In the present study, differences in microbiomes among the healthy, gingivitis, and periodontitis statuses were investigated using metatranscriptomic and network analyses. To minimize the influences of individual differences, subgingival plaque samples were collected from a healthy, a gingivitis, and a periodontitis site from each patient.

Based on 16S rRNA analysis, the read abundances of bacteria and PCoA assessments clarified differences in microbiomes between periodontitis sites and either healthy or gingivitis sites. These results suggested the presence of diseased-microbiome specificity in periodontitis sites and supported a previous finding that individual differences in microbiomes were smaller among periodontitis samples than among healthy samples (19). In periodontitis sites, high abundances of red complex bacteria and *F. nucleatum* were observed, consistent with previous findings (16, 17, 23). However, well-known putative periodontopathic bacteria were also present in healthy and gingivitis sites. Periodontopathic bacteria have been found in the microbiomes of a periodontally healthy site, although they exhibit very low abundances (23, 24). These results implied the need to consider whether such bacteria are simply present or functionally active in healthy sites. Accordingly, we assessed bacterial activities based on mRNA/16S rRNA read abundances; we found that red complex bacteria demonstrated increased activity in conjunction with disease progression. Furthermore, taxa such as *F. fastidiosum*, *E. nodatum*, *F. alocis*, and *E. saphenum* showed greater rates of mRNA/16S rRNA read abundance enhancement than did red complex bacteria, in conjunction with disease progression. We presumed that, during the development of periodontal diseases, these species might be keystone species (25, 26) and/or inflammophilic pathobionts (27, 28).

To clarify interactions among the bacteria in each microbiome, we performed co-occurrence network analyses. Compared with networks in healthy and gingivitis sites, the bacterial network in periodontitis sites had fewer taxa and showed smaller clustering coefficient values, indicating that the selected taxa engage in dysbiosis, which con-

**FIG 2** Legend (Continued)

SE relative abundances. *, significant difference between gingivitis and periodontitis sites; †, significant difference between healthy and periodontitis sites; §, significant difference between healthy and gingivitis sites. (B) PCoA plot prepared from mRNA profiles assigned via the SEED subsystem analysis, as described in the legend of Fig. 1B. (C) PCoA plot prepared from mRNA profiles assigned via the KEGG database analyses, as described in the legend of Fig. 1B. (D) Active KEGG pathways present in any of the 21 samples for each periodontal status (left maps) and status-specific pathways detected in ≥11 samples (right maps), with corresponding colors in the box below.

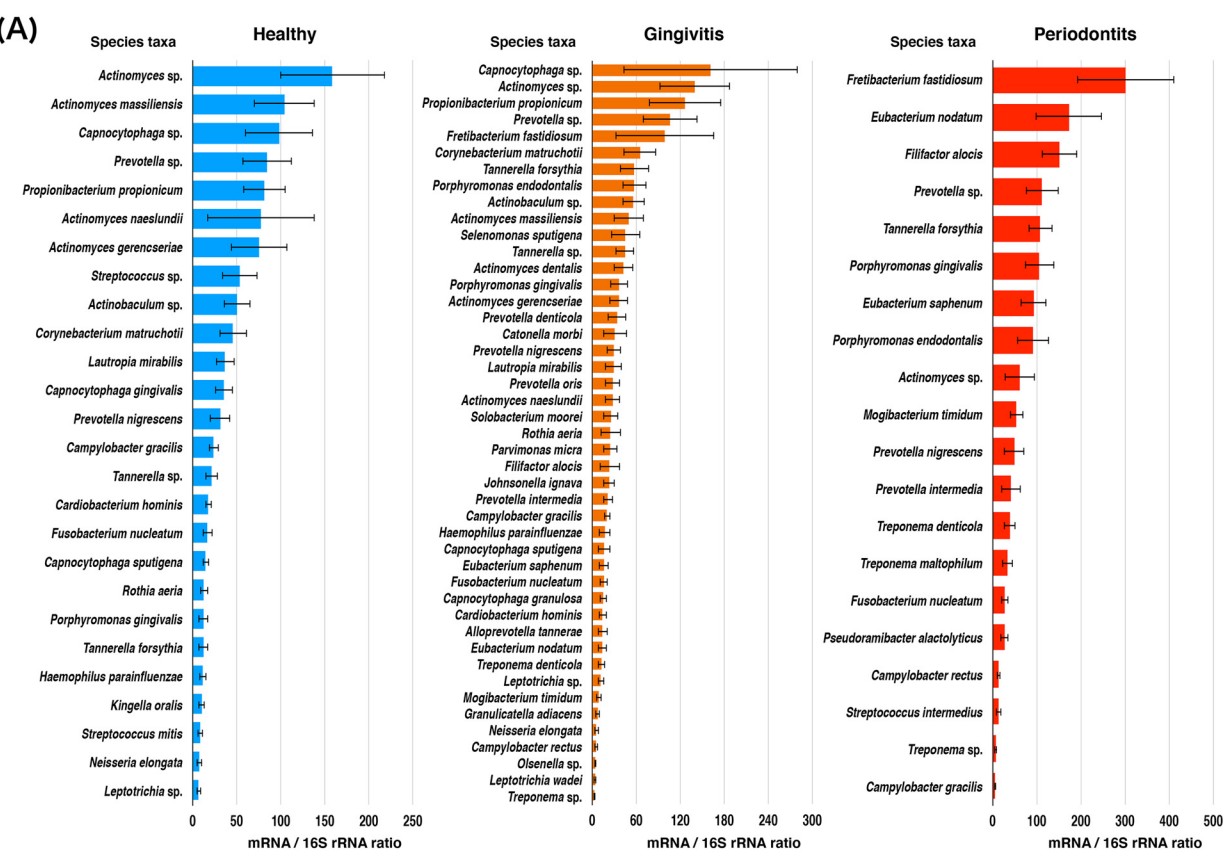

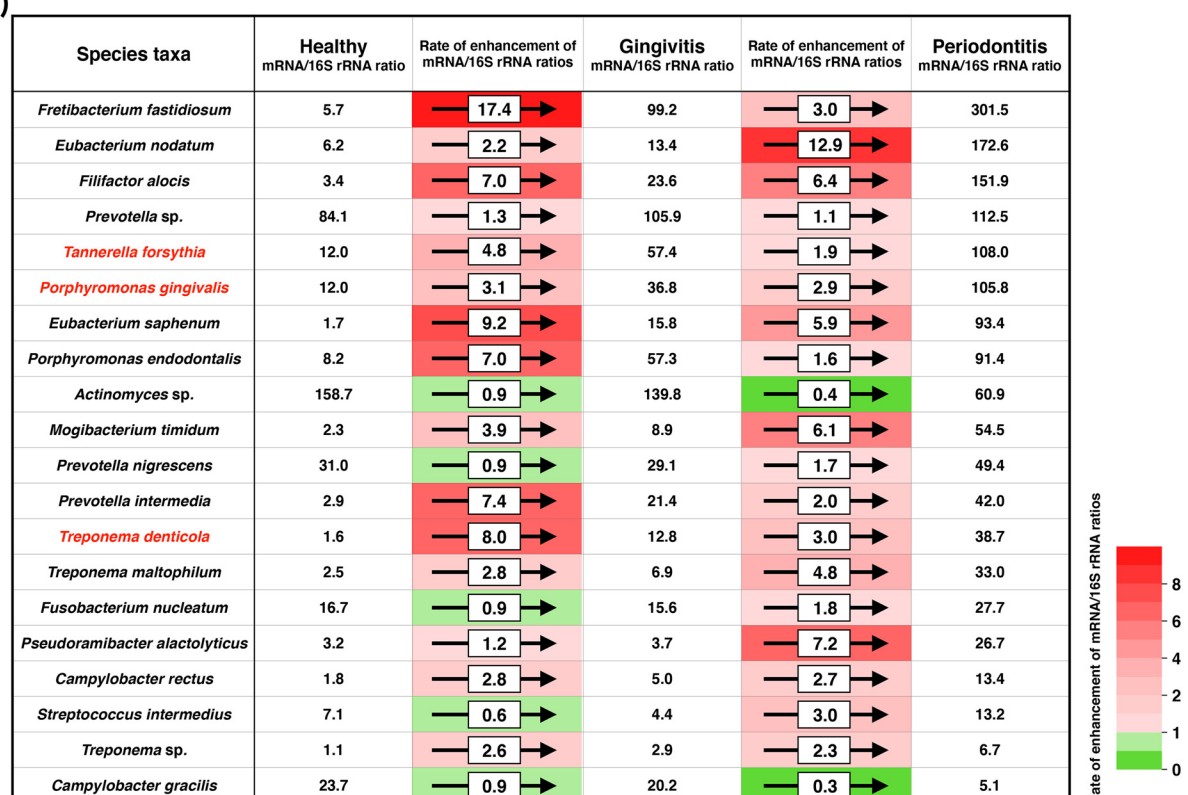

**FIG 3** Comparison of taxonomic profiles of 16S rRNA and taxonomic mRNA. (A) mRNA/rc-rRNA abundance ratios calculated for each VTiF; predominant taxa with significant differences between rc-rRNA and mRNA are shown in descending order. The bars show mean ± SE

tributes to the establishment of periodontitis. *P. nigrescens*, an interacting core taxon in periodontitis, was associated with both healthy and diseased periodontal sites (29–32). Szafrański et al. suggested that this bacterium may become an accessory pathogen in a dysbiotic community (23). *P. nigrescens* reportedly possesses a virulence-related gene, Gingipain R1 (20), which encodes a cysteine protease that serves as a primary virulence factor for *P. gingivalis*. Our findings suggest that *P. nigrescens* plays an important role in the pathogenesis of periodontitis. Regarding the gingivitis microbiome, the co-occurrence network included bacterial features of healthy and periodontitis sites; this finding was consistent with the results of PCoA. The microbiome associated with gingivitis suggested that gingivitis represents a transitional stage between health and periodontitis. The interacting core taxa of subgroup 1 in the gingivitis microbiome comprised *E. nodatum*, *E. saphenum*, *F. alocis*, *F. fastidiosum*, *M. timidum*, and *T. denticola*. As mentioned above, the rates of mRNA/16S rRNA read abundance enhancements of these taxa were highly varied, in conjunction with periodontal disease progression; thus, we presumed that they were important bacteria involved in the pathogenesis of periodontal disease. Immune inactivation activities of *E. nodatum*, *E. saphenum*, *F. alocis*, *Parvimonas micra*, and *Porphyromonas endodontalis* have been reported (33); co-occurrence relationships among these taxa were also observed in the present study. Additionally, *F. alocis* has been reported to invade epithelial tissues and substantially affect the formation of periodontal microorganism communities (34). Interactions among *E. nodatum*, *E. saphenum*, *F. alocis*, *P. micra*, and *P. endodontalis* presumably modulate the host immune response; in this study, the co-occurrence relationships among these taxa were associated with the progression of gingival inflammation. The higher mRNA expression levels of genes from these bacteria might represent characteristics of the gingivitis microbiome, while indicating a future microbiome shift toward periodontitis; therefore, monitoring these bacteria might be helpful for evaluation of disease status. In the present study, red complex bacteria exhibited unexpected behaviors in bacterial networks. All red complex bacteria were present in both the gingivitis and periodontitis networks, while co-occurrence relationships of all three species were observed only in the gingivitis network. Furthermore, *P. gingivalis* and *T. forsythia* were interacting core taxa in healthy and gingivitis sites, indicating that these bacteria were both present and functionally active in healthy and gingivitis sites. Previous studies showed that these bacteria affected primary polymicrobial biofilm formation (35). Our results suggest that interspecies interactions between red complex bacteria and other taxa are essential for the establishment of a dysbiotic community, while the roles of red complex bacteria as periodontal pathogens diminish during subsequent stable disease.

In this study, status-specific metabolic pathways were detected by the KEGG database analysis. For example, the inositol phosphate metabolism and nitrogen metabolism pathways were found to be active only in H and G sites. A recent study of metabolomic profiles in periodontitis showed that inositol levels of gingival crevicular fluid were significantly higher in periodontally healthy subjects than in patients with periodontitis (36). Furthermore, Hoffman et al. noted that nitrogen metabolism prevailed in a *Prevotella*-dominant bacterial community (37). We also found a high rate of *Prevotella intermedia* mRNA/16S rRNA ratios between H and G sites, which is consistent with the previous results. However, the relationships between the other active pathways and periodontal bacteria that were present in each status need further clarification. Analyses based on the SEED subsystems and the KEGG database indicated that bacterial chemotaxis and flagellar assembly were enriched in conjunction with periodontal disease progression. Duran-Pinedo et al. and Yost et al. considered that increases in these metabolic activities were

**FIG 3** Legend (Continued)

mRNA/rc-rRNA abundance ratios. (B) Comparison of mRNA/16S rRNA ratios between healthy and gingivitis sites and between gingivitis and periodontitis sites; rates of enhancement of mRNA/16S rRNA ratios for each taxon are depicted, showing significant differences in read abundance between rc-rRNA and mRNA in periodontitis sites. mRNA/16S rRNA ratios for each taxon are also depicted. Rates of enhancement of mRNA/16S rRNA ratios are indicated by color gradient. Red complex taxa are indicated in a red font.

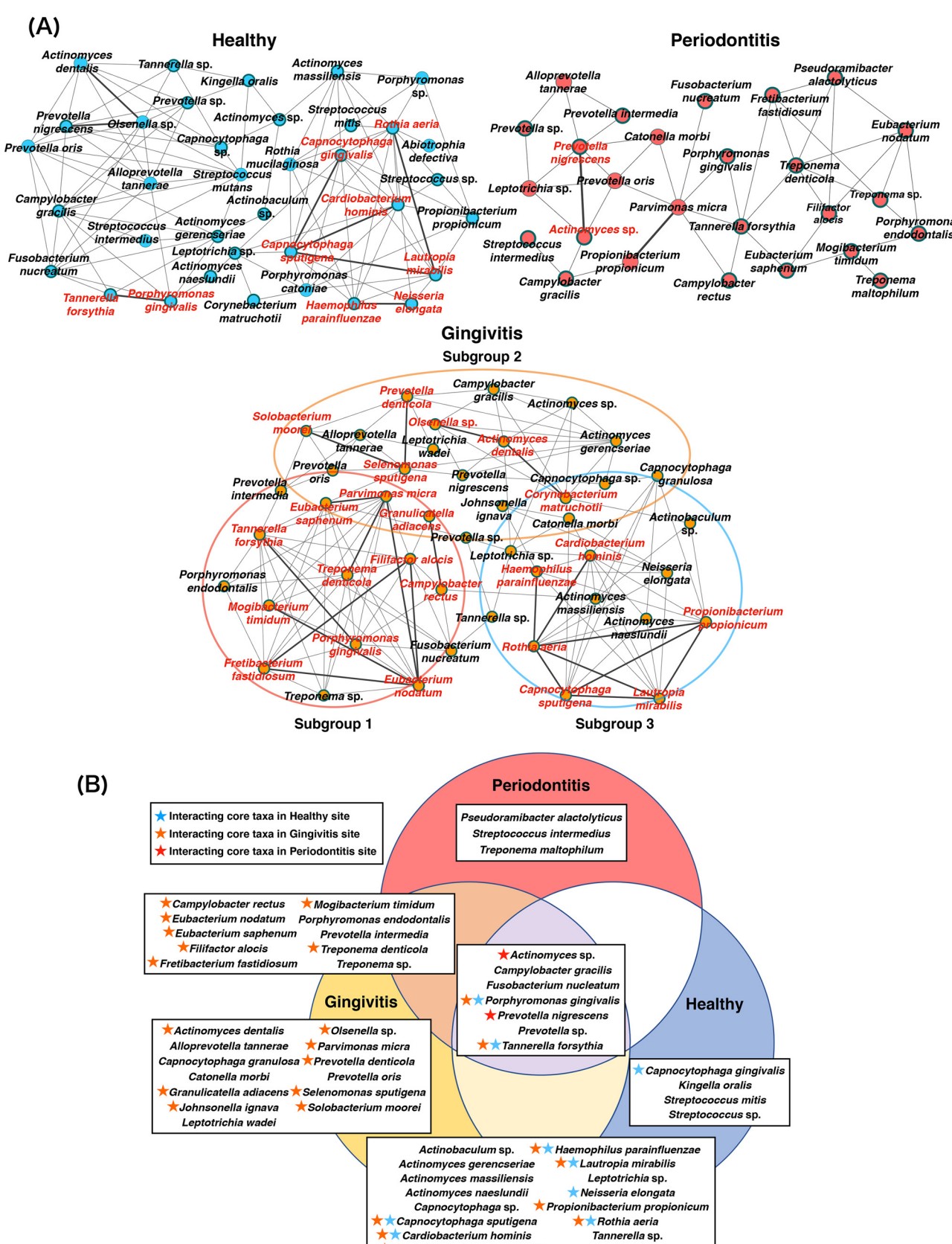

**FIG 4** Co-occurrence networks and interacting core taxa in VTiF profiles. (A) All networks are shown, with each bacterial taxon and co-occurrence relationship indicated by a node and an edge, respectively. Active taxa are indicated with bold circles, and interactions with significant co-occurrence

significant in defining severe periodontitis (16, 38). Furthermore, the genes involved in these metabolic activities were upregulated in the context of periodontitis, in a manner driven by *F. alocis*, *F. fastidiosum*, and red complex bacteria (39). In the present study, rates of mRNA/16S rRNA read abundance enhancement and interactions of co-occurrence networks related to these bacteria were observed in the gingivitis and periodontitis microbiomes. Additionally, several types of amino acid metabolism (e.g., glutamate and aspartate metabolism) were upregulated in conjunction with periodontal disease progression. Previous metatranscriptomic analyses of the periodontal microbiome suggested that amino acid metabolism tended to be enriched in periodontitis sites (38, 40). In addition, fatty acid biosynthesis was presumably related to the presence of red complex bacteria and the onset of periodontal disease (41). These findings suggest that the stricter anaerobic environment in deep periodontal pockets is more suitable for anaerobe-predominant microbiomes and the expression of genes within anaerobic metabolic pathways (e.g., fermentation and fatty acid biosynthesis). However, robust expression patterns of genes related to aerobic metabolism (e.g., ATP synthases and sugar-related metabolism) were observed in both healthy and gingivitis sites, implying similar surrounding environments. These results are supported by the similarity of co-occurrence networks in healthy and gingivitis sites in this study; more taxa were commonly detected and the clustering coefficient values were greater in both healthy and gingivitis sites than in periodontitis sites. These findings imply an invertible transition between the healthy and gingivitis statuses.

A notable limitation of this study is that the microbiological analyses were conducted with a limited amount of cross-sectional microbiological data. Longitudinal investigations of microbiomes and their clinical relationships with a larger number of samples are needed to elucidate bacterial changes during the health-to-periodontitis shift and confirm the findings reported in this study. Also, we should recognize that the metatranscriptome does not necessarily represent the final metabolic products generated by the microbial community. Future studies using omics approaches are necessary to clarify the bacteria and/or virulence factors that are truly associated with the development of periodontal diseases.

In conclusion, we elucidated differences in bacterial compositions and gene expression profiles among the healthy, gingivitis, and periodontitis statuses. Co-occurrence network analysis revealed that the gingivitis network included features of networks observed in healthy and periodontitis sites. In particular, interacting core taxa in the gingivitis microbiome (e.g., *E. nodatum*, *E. saphenum*, *F. alocis*, *F. fastidiosum*, *M. timidum*, and *T. denticola*) may play important roles and offer bacterial targets for evaluation of disease progression.

## MATERIALS AND METHODS

**Ethical statement.** This study was performed in accordance with the Ethical Guidelines for Clinical Studies (2008 notification number 415 of the Ministry of Health, Labor, and Welfare) and was approved by the Ethics Committee of Tokyo Medical and Dental University (D2015-535). All patients provided written, informed consent prior to participating in this study. The study was conducted in accordance with the principles of the Declaration of Helsinki, as revised in 2013.

**Study population.** Twenty-one patients seeking dental treatment in the Dental Hospital of Tokyo Medical and Dental University were recruited for this study. The patients had healthy (PD ≤ 3 mm without bleeding on probing [BOP]), gingivitis (PD ≤ 3 mm with BOP), and periodontitis (PD ≥ 4 mm with BOP, clinical attachment loss, and radiographic bone loss) sites in maxillary or mandibular anterior teeth (42, 43). These patients were systemically healthy and had not received systemic antibiotics or anti-inflammatory agents within 3 months prior to the start of this study (17, 44). Clinical periodontal parameters (PD and BOP) were measured at six sites per tooth: mesiobuccal, buccal, distobuccal, mesiolingual, lingual, and distolingual sites.

**Sample collection and RNA extraction.** Subgingival plaque samples were obtained from the deepest pocket with or without BOP in the healthy, gingivitis, and periodontitis sites, respectively (three sin-

**FIG 4** Legend (Continued)

are indicated with bold lines. Interacting core taxa are indicated in red text for each status. The gingivitis microbiome network is presented as three subgroups according to their characteristics. (B) VTiF with significant differences between rc-rRNA and mRNA. Interacting core taxa are shown with star symbols (red, taxa constituted interacting core taxa in periodontitis sites; orange, taxa constituted interacting core taxa in gingivitis sites; blue, taxa constituted interacting core taxa in healthy sites).

**TABLE 2** Interactions with significant co-occurrence

| Site specificity | Species taxon | mRNA/16S rRNA ratio | No. of samples detected | Species taxon | mRNA/16S rRNA ratio | No. of samples detected | Positive correlation coefficient |
|---|---|---|---|---|---|---|---|
| Healthy | Capnocytophaga sputigena | 21.407 | 12 | Capnocytophaga gingivalis | 62.871 | 12 | 0.805 |
| | Cardiobacterium hominis | 20.067 | 16 | Capnocytophaga sputigena | 21.407 | 12 | 0.692 |
| | Lautropia mirabilis | 35.117 | 14 | Capnocytophaga sputigena | 21.407 | 12 | 0.710 |
| | Neisseria elongata | 13.623 | 8 | Haemophilus parainfluenzae | 22.465 | 9 | 0.771 |
| | Olsenella sp. | 9.679 | 6 | Actinomyces dentalis | 50.461 | 6 | 0.791 |
| | Prevotella nigrescens | 49.652 | 8 | Olsenella sp. | 9.679 | 6 | 0.739 |
| | Rothia aeria | 13.698 | 8 | Lautropia mirabilis | 35.117 | 14 | 0.713 |
| | Tannerella forsythia | 25.074 | 7 | Porphyromonas gingivalis | 20.982 | 7 | 0.852 |
| Gingivitis | Corynebacterium matruchotii | 65.617 | 12 | Actinomyces dentalis | 72.599 | 10 | 0.690 |
| | Filifactor alocis | 29.357 | 6 | Eubacterium nodatum | 12.626 | 6 | 0.565 |
| | Fretibacterium fastidiosum | 146.576 | 6 | Eubacterium nodatum | 12.626 | 6 | 0.621 |
| | Fretibacterium fastidiosum | 146.576 | 6 | Filifactor alocis | 29.357 | 6 | 0.566 |
| | Granulicatella adiacens | 6.846 | 6 | Campylobacter rectus | 7.757 | 12 | 0.526 |
| | Lautropia mirabilis | 29.974 | 9 | Capnocytophaga sputigena | 12.844 | 8 | 0.726 |
| | Mogibacterium timidum | 16.397 | 7 | Eubacterium nodatum | 12.626 | 6 | 0.629 |
| | Olsenella sp. | 7.505 | 9 | Actinomyces dentalis | 72.599 | 10 | 0.818 |
| | Parvimonas micra | 69.657 | 6 | Eubacterium nodatum | 12.626 | 6 | 0.726 |
| | Parvimonas micra | 69.657 | 6 | Eubacterium saphenum | 32.043 | 6 | 0.647 |
| | Propionibacterium propionicum | 145.588 | 13 | Capnocytophaga sputigena | 12.844 | 8 | 0.612 |
| | Propionibacterium propionicum | 145.588 | 13 | Lautropia mirabilis | 29.974 | 9 | 0.749 |
| | Rothia aeria | 65.440 | 6 | Capnocytophaga sputigena | 12.844 | 8 | 0.624 |
| | Rothia aeria | 65.440 | 6 | Cardiobacterium hominis | 18.214 | 11 | 0.669 |
| | Rothia aeria | 65.440 | 6 | Haemophilus parainfluenzae | 39.447 | 7 | 0.578 |
| | Rothia aeria | 65.440 | 6 | Lautropia mirabilis | 29.974 | 9 | 0.849 |
| | Rothia aeria | 65.440 | 6 | Propionibacterium propionicum | 145.588 | 13 | 0.645 |
| | Selenomonas sputigena | 173.412 | 6 | Prevotella denticola | 71.881 | 7 | 0.820 |
| | Solobacterium moorei | 101.266 | 6 | Selenomonas sputigena | 173.412 | 6 | 0.622 |
| | Tannerella forsythia | 13.854 | 15 | Porphyromonas gingivalis | 11.045 | 11 | 0.690 |
| | Treponema denticola | 10.396 | 9 | Parvimonas micra | 69.657 | 6 | 0.690 |
| Periodontitis | Prevotella nigrescens | 52.208 | 8 | Actinomyces sp. | 20.084 | 7 | 0.705 |
| | Propionibacterium propionicum | 37.131 | 6 | Parvimonas micra | 79.772 | 6 | 0.664 |

gle sites per patient). The sampling sites were isolated with sterile cotton rolls, and supragingival plaque was removed by sterile cotton pellets. The sites were dried by air spray, and 10 sterilized paper points were inserted into the pocket for 60 s each. The points were then collected in a sterilized tube and stored at −80°C until use.

RNA was extracted using the PowerMicrobiome RNA isolation kit (MO BIO Laboratories, Carlsbad, CA, USA) and purified using the NucleoSpin miRNA kit (Clontech, Mountain View, CA, USA), the Dr. GenTLE precipitation carrier (TaKaRa Bio, Shiga, Japan), and TURBO DNase (Ambion, Austin, TX, USA), in accordance with a published method (17, 18). Purified RNA was quantified using a Quantus fluorometer (Promega, Madison, WI, USA), and RNA quality was evaluated by capillary electrophoresis with an Agilent 2100 bioanalyzer (Agilent Technologies, Santa Clara, CA, USA).

**cDNA synthesis, library preparation, and Illumina sequencing.** Purified RNA was polyadenylated using the A-Plus poly(A) polymerase tailing kit (Epicentre, Madison, WI, USA) in accordance with the manufacturer's protocol and then concentrated by ethanol precipitation using the Dr. GenTLE precipitation carrier. The polyadenylated RNA was reverse transcribed into cDNA; 15 cycles of amplification were conducted using the SMART-Seq v4 ultralow-input RNA kit for sequencing (TaKaRa Bio), in accordance with the manufacturer's protocol. Sequencing libraries were prepared with the Nextera XT DNA sample preparation kit (Illumina, San Diego, CA, USA), in accordance with the manufacturer's protocol. Amplified cDNA was quantified by real-time PCR on a LightCycler (Roche Diagnostics, Mannheim, Germany) with the KAPA library quantification kit of Illumina (KAPA Biosystems, Wilmington, MA, USA), in accordance with the manufacturer's protocol; DNA quality was evaluated by capillary electrophoresis with an Agilent 2100 bioanalyzer. Prepared samples from 21 patients (63 samples in total) were pooled, and the Illumina MiSeq platform was used to generate 300-bp paired-end reads.

**Processing and analyzing of Illumina sequencing data.** Processing and analysis of Illumina sequencing data (see Table S4 in the supplemental material) were performed in accordance with a published protocol (17, 18). Briefly, the data were initially processed using Trimmomatic software, version 0.32 (45), for quality trimming and adapter clipping. In addition, sequences of human origin (9,196,859 total reads; mean proportion, 21.1%) were removed using DeconSeq software, version 0.4.3 (46). The data were then further processed using cmpfastq software for separation of paired and unpaired reads (47).

Only paired reads were used for 16S rRNA analysis and OTU identification by expectation maximization iterative reconstruction of genes from the environment (EMIRGE) (48). The number of reads in each rc-rRNA OTU was calculated as the abundance value of each 16S rRNA OTU. The representative sequence of each 16S rRNA OTU was aligned for a nucleotide similarity search against sequences in the Human Oral Microbiome Database, version 13.2 (49), using Basic Local Alignment Search Tool N (BLASTN) (https://blast.ncbi.nlm.nih.gov/Blast.cgi?PAGE_TYPE=BlastSearch). The abundance values of all 16S rRNA OTUs were normalized by conversion to reads per kilobase of transcript per million reads (RPKM). The parameters used for these tools were established in accordance with a previous study (17, 18).

The community diversities of all samples were estimated by comparing complexities among healthy, gingivitis, and periodontitis sites using RPKM values. Alpha diversity indexes were estimated from the number of OTUs and the Shannon index. Rarefaction curves were drawn from abundance values before conversion into RPKM values with the rarefaction.single command in mothur software, version 1.33.3 (50). Beta diversity was estimated by PCoA using the R software package ape (https://github.com/cran/ape). Dissimilarity matrixes were generated on the basis of Bray-Curtis dissimilarity using the R software package vegan (https://github.com/vegandevs/vegan); the matrixes were used to perform PCoA in which each sample was plotted with three coordinates.

Paired and unpaired reads were annotated for mRNA analysis to confirm functional categories, metabolic pathways, and bacterial genes. The Metagenomics Rapid Annotation using Subsystem Technology pipeline (51) was used to investigate functional categories and metabolic pathways. All abundance values were normalized by conversion to reads per million reads. Bar plots were used to illustrate the compositions of mRNA profiles by assignments made with the level 1 SEED subsystems. Active pathways in the KEGG database (52) were visualized using iPath3 (53). For confirmation of bacterial genes, paired reads were merged by fastq-join (54). Furthermore, all reads, including nonmerged and unpaired reads, were formed into OTUs by Cluster Database at High Identity with Tolerance software. OTUs derived from 16S rRNA were removed by similarity comparison with BLASTN against SILVA (release 119) (55). The remaining OTUs were considered to be derived from mRNA. The mRNA OTUs were used to assign protein functions by BLASTX against the NCBI nr protein database (as of 31 October 2014). Abundance values of all mRNA OTUs were normalized by conversion to RPKM values, which were also used to determine the abundances of taxonomic origins. The parameters used for these analyses were established in accordance with those of a previous study (17, 18).

The mRNA OTUs were also used to identify putative virulence factors by using BLASTX against VFDB (as of 9 February 2015) and MvirDB (as of 9 October 2014). Differences in taxonomic profiles between 16S rRNA and mRNA OTUs were used to evaluate the activities of individual taxa. Taxa that were assigned both 16S rRNA and mRNA OTUs were listed, while the remaining taxa were excluded from this analysis. Taxa that were detected in both 16S rRNA and mRNA profiles were defined as VTiF (17, 18). To compare the abundances of 16S rRNA OTUs and mRNA OTUs, the abundance values of all 16S rRNA OTUs were normalized by conversion into RPKM values. The VTiF with mRNA/16S rRNA read abundance was regarded as an indicator of the viability and functionality of bacterial taxa responsible for disease etiology (17, 18). VTiF with an mRNA/16S rRNA read abundance of >1 were used to define active taxa. To understand detailed co-occurrence relationships in mRNA profiles of VTiF, VTiF present in at least six patients (>25% of the participants) were extracted, followed by the creation of network structures (i.e., co-occurrence networks). The co-occurrence coefficients were calculated using SparCC software (56) by

mRNA taxonomic abundances. Taxon pairs with SparCC values of ≥0.3 were regarded as positive co-occurrence relationships, and networks were visualized using Cytoscape software, version 3.7.2 (57).

**Statistical analysis.** PERMANOVA was conducted to compare species compositions and functions in microbiomes among the three periodontal statuses. One-way analysis of variance of Friedman's test with the Dunn *post hoc* test was used for comparisons of clinical parameters, alpha diversities, and read abundances of rc-rRNA OTUs and bacterial genes among the three periodontal statuses (58). ANOSIM was used to test the significance of dissimilarity among statuses in the bacterial compositions of mRNA and rc-rRNA clusters based on read abundances. Wilcoxon's signed-rank test was used to test for significant differences between mRNA and rc-rRNA read abundances in each taxon. In all statistical tests, $P$ values of $<0.05$ were considered statistically significant. All analyses were performed using R software, version 3.1.1 (R Foundation for Statistical Computing, Vienna, Austria).

**Data availability.** The data sets generated for this study can be found in the DNA Data Bank of Japan (DDBJ) with the following accession number for RNA sequencing: DRA011737.

## SUPPLEMENTAL MATERIAL

Supplemental material is available online only.

**TEXT S1**, DOCX file, 0.01 MB.
**FIG S1**, PDF file, 0.5 MB.
**FIG S2**, PDF file, 0.1 MB.
**FIG S3**, PDF file, 0.7 MB.
**FIG S4**, PDF file, 0.1 MB.
**FIG S5**, PDF file, 0.1 MB.
**TABLE S1**, XLSX file, 15.8 MB.
**TABLE S2**, XLSX file, 0.1 MB.
**TABLE S3**, XLSX file, 0.04 MB.
**TABLE S4**, XLSX file, 0.03 MB.

## ACKNOWLEDGMENTS

This work was supported by the Japan Society for the Promotion of Science (grant 19K10164 to T.K., grants 17H06662, 19K19016, and 21K16987 to T.S., and grants 17K11981 and 20K09934 to Y.T.). Supercomputing resources were provided by the Human Genome Center at the Institute of Medical Science (University of Tokyo) (http://sc.hgc.jp/shirokane.html).

We thank Hugo Song from HUGO LS (https://www.hugols.com) and Ryan Chastain-Gross from Edanz (https://jp.edanz.com/ac) for editing a draft of the manuscript.

T. Nemoto, T.S., and Y.T. performed the experiments, processed the sequence data, and wrote the first draft of the manuscript. K.K., T.W., M.S., M.S., T.K., and T. Nagai assisted with the experiments and reviewed the manuscript. T.S., S.K., Y.T., and T.I. supervised the analyses and wrote the manuscript. All authors read and approved the final version of the manuscript.

We declare that the research was conducted in the absence of any commercial or financial relationships that could be construed as a potential conflict of interest.

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
