## [Reviewer comments · mSystems]

Discrimination of Bacterial Community Structure among Healthy, Gingivitis, and Periodontitis Statuses through Integrated Metatranscriptomic and Network Analyses

Takashi Nemoto, Takahiko Shiba, Keiji Komatsu, Takayasu Watanabe, Masahiro Shimogishi, Masaki Shibasaki, Tatsuro Koyanagi, Takahiko Nagai, Sayaka Katagiri, Yasuo Takeuchi, and Takanori Iwata

Corresponding Author(s): Yasuo Takeuchi, Tokyo Medical and Dental University

Review Timeline:

Submission Date:	July 9, 2021
Editorial Decision:	August 23, 2021
Revision Received:	September 16, 2021
Accepted:	September 20, 2021

Editor: Holly Bik

Reviewer(s): Disclosure of reviewer identity is with reference to reviewer comments included in decision letter(s). The following individuals involved in review of your submission have agreed to reveal their identity: Gena D Tribble (Reviewer #2)

Transaction Report:

DOI: <https://doi.org/10.1128/mSystems.00886-21>

August 23, 2021

Dr. Yasuo Takeuchi
Tokyo Medical and Dental University
Department of Periodontology, Graduate School of Medical and Dental Sciences
Tokyo
Japan

Re: mSystems00886-21 (Discrimination of Bacterial Community Structure among Healthy, Gingivitis, and Periodontitis Statuses through Integrated Metatranscriptomic and Network Analyses)

Dear Dr. Yasuo Takeuchi:

Thank you for submitting your manuscript to mSystems. We have completed our review and I am pleased to inform you that, in principle, we expect to accept it for publication in mSystems. However, acceptance will not be final until you have adequately addressed the reviewer comments.

The two reviewers were very enthusiastic about the scope of this manuscript and its findings, however they have a number of comments highlighting areas where this submission could be improved. In addition to addressing each reviewer comment, the authors also need to address the following points:

- Ensure that the sequence data is made publicly available. I could not find a record for the accession number DRA011737 on the DDBJ website.
- This manuscript has a large number of supplemental files/figures. Please reduce this to <10 supplemental items as per journal guidelines.
- Figures should include error bars when presenting results averaged from a group. The authors should also more clearly display the number of samples on the figures.

Preparing Revision Guidelines

For complete guidelines on revision requirements, please see the journal Submission and Review Process requirements at <https://journals.asm.org/journal/mSystems/submission-review-process>.
Submission of a paper that does not conform to mSystems guidelines will delay acceptance of your manuscript.

Sincerely,

Holly Bik

Editor, mSystems

Journals Department
Reviewer comments:

Reviewer #1 (Comments for the Author):

This is a timely study that aimed to investigate the bacterial characteristics of healthy, gingivitis, and periodontitis statuses through metatranscriptomic analysis. Subgingival plaque samples of healthy, gingivitis, and periodontitis sites in the same oral cavity were collected from 21 patients. 16S rRNA analysis was used to define bacterial composition, whereas the taxonomic and functional profiles of the communities were characterized by mRNA analysis. The results indicate differences in bacterial compositions and functional profiles between healthy and periodontitis sites. Co-occurrence networks were constructed for each group by connecting two bacterial species if their mRNA abundances were positively correlated. The clustering coefficient analysis revealed that the (species-based) co-occurrence network complexity increased during gingivitis development, but it decreased during progression to periodontitis. Notable species displaying greater transcriptional activities associated with disease progression were *Eubacterium nodatum*, *Eubacterium saphenum*, *Filifactor alocis*, and *Fretibacterium fastidiosum*, in contrast to the 'red complex' bacteria. The study highlights the gingivitis structural and functional microbiome profile as an important transitory stage from health to periodontitis.

This is an interesting perspective with a well justified study approach. The comparison of taxonomic profiles of 16S rRNA and taxonomic mRNA (eg. comparison of mRNA/16S rRNA ratios between clinical states) are particularly important, in determining the significantly active taxa. Some points

are raised after reviewing if the manuscript.

The English language text needs some attention for correction of some grammatical errors.

1. Recent review work on metatranscriptomics (PMID: 33226688) and metagenomics of the oral microbiome (PMID: PMID: 33226714) should be acknowledged, and the collective findings should be compared to the present findings.
2. In lines 156-157, the word "expressed" should be introduced "...1,575 bacterial genes were commonly expressed among all three periodontal statuses ..."
3. In lines 183-188 it is reported that five pathways were active only in gingivitis and periodontitis sites, whereas five other pathways were active only in healthy and gingivitis sites. These findings need to be elaborated further in the Discussion section.
4. In lines 264-266 it is stated that "Based on the keystone pathogen theory (24, 25), we presumed that these species 265 might constitute inflammophilic pathobionts and/or keystone species in periodontal diseases". However, the keystone pathogen theory is not commensurate with the inflammophilic pathobiont phenotypic profile of the species, and therefore these terms need to be distinguished from one another.
5. In lines 330-331 it is stated that: "These findings imply an easy transition between healthy and gingivitis statuses". The term "easy" is not well comprehended here, so the authors may choose an alternative one.
6. A limitation of the study as noted in line 331 is its cross-sectional nature, and that future longitudinal investigations of microbiomes and their clinical relationships will elucidate further the bacterial changes during the conversion of health to periodontitis. I would also note that the number of participants is rather limited for this type of study is rather limited and recommend it include this among the limitations.

Reviewer #2 (Comments for the Author):

This is an excellent study utilizing transcriptomics to define bacterial communities in three common stages of periodontal disease. Advantages of the study include sampling within subjects for all three disease states, and generating ratios of gene expression over bacterial counts to create a novel view of bacterial viability/activity and interactions. I have some comments to improve readability for non-dental microbiologists, but otherwise I have no major concerns and find this study to be a valuable contribution to the literature.

Topic: comparing communities of health-gingivitis-perio by gene expression within subjects And network linkage based on gene expression. The authors have previous publications using similar approaches and the writing is generally very clear.

-The authors repeatedly indicate that the composition of the oral microbiome in health, gingivitis, and periodontitis is not well studied, but that is far from the case. The utility of this particular work is the profiling based on levels of gene expression rather than exclusively bacterial numbers. I think the authors need to consider the following list of comments and adjust the emphasis accordingly. **The novelty and value in the study is in defining the communities by transcriptomics.**

Line 32-35

'However, there remains an inadequate understanding regarding bacteria that are depleted or enriched during the health-to-periodontitis microbiome shift, as well as bacteria associated with gingivitis and/or periodontitis.'

This statement is not strictly accurate, as the presence/absence is understood, however the METABOLIC ACTIVITY during these stages is not, so I suggest modifying this line accordingly.

Line 59 regarding the difference in microbiome and **it's functional activity in** healthy, gingivitis, and periodontitis

Line 65-67

These results provide ~~bacteriological~~ transcriptomic evidence to support gingivitis as an intermediate state between healthy and periodontitis statuses.

-mSystems is not a dental-oriented journal, and the target audience will need careful explanation of dental terms. In that regard, I suggest the following adjustments to the text.

Line 74. Add an additional line defining the periodontium. "The periodontium is comprised of the soft tissue and bone surrounding the tooth, *and periodontal diseases are representative ~~oral~~ polymicrobial diseases, which involve a microbiome imbalance known as dysbiosis that triggers periodontal inflammation (1, 2).*

Line 79. *There is increasing evidence that periodontal disease leads to tooth loss and oral functional decline, as well as the onset or progression of various systemic diseases (5-9). We know that periodontitis leads to tooth loss and oral functional decline, thus I suggest the following adjustment to the wording. Periodontitis leads to tooth loss and oral functional decline, and there is increasing evidence that it is associated with onset or progression of various systemic diseases (5-9).*

Other comments:

Line 501 : Table 1 legend. Include the information that age and standard deviation are in years.

Line 130-131 Table S1 shows data at the species level, not genus level as described in the text, which is a bit confusing. Please explain or modify.

Lines 123-125 These total numbers of reconstructed OTUs seems low compared to DNA-based methods. Are these values consistent with other transcriptomics studies?

Line 198-199 *The total number of taxa in H, G, and P sites were 2,058, 2,279, and 2,129, respectively;* These seems like an extremely high number, please describe briefly how taxa are defined here (at the genus level, species level, subspecies level, or a mix?)

Lines 210 -211. *Among active taxa, the activities of eight taxa (*F. fastidiosum*, *Eubacterium nodatum*, *F. alocis*, *Actinomyces sp.*, *Prevotella sp.*, *E. saphenum*, *Porphyromonas endodontalis*, and *P. nigrescens*) were greater than the activities of red complex bacteria in P site.* Comparing this text to figure 3AB, only *F. fastidiosum*, *Eubacterium nodatum*, *F. alocis*, and *Prevotella sp* have a higher mRNA/16srRNA ratio than two of the red species complex. This statement should be modified.

Line 225-227 *The significant active taxa (26/36, 45/45, and 20/26 taxa in H, G, and P sites, respectively) were prevalent in all networks (Figure 4A, Table 2, and 227 Tables S6–S8).* I am not sure I understand this statement correctly. The input data for correlation analysis was the ratio data, so wouldn't you expect the highly active taxa to be present in the network? And they were prevalent in their own network, but not ALL networks? For example, there are 45 taxa in the G network, but they are not all present in the H or P networks? This will require additional clarification in the text.

Line 240. *Approximately half of adults over 30 years of age exhibit periodontal disease in North America.* While this is probably true, the study was done in Japanese adults and in fact periodontal disease is common world-wide. Therefore this is not the best opening statement for the discussion, in my opinion, maybe something about occurrence world-wide is better.

**Editor**

The two reviewers were very enthusiastic about the scope of this manuscript and
findings, however they have a number of comments highlighting areas where this
submission could be improved. In addition to addressing each reviewer comment, the
authors also need to address the following points.

**Response:** We thank the reviewers and editor for their insightful comments, which have
helped us significantly improve our manuscript.

**1. Ensure that the sequence data is made publicly available. I could not find a**
**record for the accession number DRA011737 on the DDBJ website.**

**Response:** We had submitted the sequence data and obtained the accession number but
had not opened it to the public. We have now made the sequence data publicly
available.

**2. This manuscript has a large number of supplemental files/figures. Please reduce**
**this to <10 supplemental items as per journal guidelines.**

**Response:** We combined some of the supplementary tables, which reduced the number
of supplemental items to 10.

**3. Figures should include error bars when presenting results averaged from a**
**group. The authors should also more clearly display the number of samples on the**
**figures.**

**Response:** We have added error bars (standard error) in Figures 1A, 2A, 3A, and S3,
and included sample number information in Figures 1B, 2B, 2C, S4, and S5.

**Reviewer #1**

This is an interesting perspective with a well justified study approach. The comparison
of taxonomic profiles of 16S rRNA and taxonomic mRNA (eg. comparison of
mRNA/16S rRNA ratios between clinical states) are particularly important, in
determining the significantly active taxa. Some points are raised after reviewing if the
manuscript. The English language text needs some attention for correction of some
grammatical errors.

**Response:** The English grammar and style were checked and revised by a native

speaker and a professional scientific editing company.

**1. Recent review work on metatranscriptomics (PMID: 33226688) and**
**metagenomics of the oral microbiome (PMID: PMID: 33226714) should be**
**acknowledged, and the collective findings should be compared to the present**
**findings.**

**Response:** We agree that the explanation of background and discussion of our results
were insufficient in the original manuscript. We have added more details in the
Introduction and Discussion sections and cited additional references (p. 4, lines 85–87;
p. 5, lines 101–105; p. 10-11, lines 240–243).

**2. In lines 156-157, the word "expressed" should be introduced "...1,575 bacterial**
**genes were commonly expressed among all three periodontal statuses ..."**

**Response:** We changed the sentence as suggested (p. 7, line 156).

**3. In lines 183-188 it is reported that five pathways were active only in gingivitis**
**and periodontitis sites, whereas five other pathways were active only in healthy**
**and gingivitis sites. These findings need to be elaborated further in the Discussion**
**section.**

**Response:** We have elaborated on the results of the pathway analysis in the Discussion
section, as suggested, and cited additional references (p. 13–14, lines 307–317).

**4. In lines 264-266 it is stated that "Based on the keystone pathogen theory (24, 25),**
**we presumed that these species 265 might constitute inflammophilic pathobionts**
**and/or keystone species in periodontal diseases". However, the keystone pathogen**
**theory is not commensurate with the inflammophilic pathobiont phenotypic profile**
**of the species, and therefore these terms need to be distinguished from one another.**

**Response:** We agree that the original wording was confusing. We have revised the
sentence as: "We presumed that, during the development of periodontal diseases, these
species might be keystone species and/or inflammophilic pathobionts." (p. 12, lines
265–266).

**5. In lines 330-331 it is stated that: "These findings imply an easy transition**
**between healthy and gingivitis statuses". The term "easy" is not well**
**comprehended here, so the authors may choose an alternative one.**

**Response:** We checked the sentence and decided to remove the word "easy" (p. 15, line

340).

**6. A limitation of the study as noted in line 331 is its cross-sectional nature, and**
**that future longitudinal investigations of microbiomes and their clinical**
**relationships will elucidate further the bacterial changes during the conversion of**
**health to periodontitis. I would also note that the number of participants is rather**
**limited for this type of study is rather limited and recommend it include this**
**among the limitations.**

**Response:** We have revised the description about the limitations and included the
limited amount of cross-sectional microbiological data as one of the limitations (p. 15,
lines 341–349).

**Reviewer #2**

The authors repeatedly indicate that the composition of the oral microbiome in health,
gingivitis, and periodontitis is not well studied, but that is far from the case. The utility
of this particular work is the profiling based on levels of gene expression rather than
exclusively bacterial numbers. I think the authors need to consider the following list of
comments and adjust the emphasis accordingly. The novelty and value in the study is in
defining the communities by transcriptomics.

**Response:** We carefully revised the manuscript according to your valuable feedback.

**1. Line 32-35 ‘However, there remains an inadequate understanding regarding**
**bacteria that are depleted or enriched during the health-to-periodontitis**
**microbiome shift, as well as bacteria associated with gingivitis and/or periodontitis.’**
**This statement is not strictly accurate, as the presence/absence is understood,**
**however the METABOLIC ACTIVITY during these stages is not, so I suggest**
**modifying this line accordingly.**

**Response:** We agree that the original sentence was inaccurate. We have revised the
sentences to better reflect the purpose and significance of our study (p. 2, lines 32–34).

**2. Line 59 regarding the difference in microbiome and *it’s functional activity in***
**healthy, gingivitis, and periodontitis**

**Response:** We have changed “it’s” to “its” in the sentence and revised the sentence (p. 3,
lines 57–58).

**3. Line 65-67 These results provide ~~bacteriological~~ transcriptomic evidence to**
**support gingivitis as an intermediate state between healthy and periodontitis statuses.**

**Response:** In accordance with this comment, we have revised the sentence (p. 3, lines
64–65).

**4. mSystems is not a dental-oriented journal, and the target audience will need**
**careful explanation of dental terms. In that regard, I suggest the following**
**adjustments to the text.**

Line 74. Add an additional line defining the periodontium. “The periodontium is
comprised of the soft tissue *and bone surrounding the tooth, and periodontal diseases*
*are representative ~~oral~~-polymicrobial diseases, which involve a microbiome imbalance*
*known as dysbiosis that triggers periodontal inflammation (1, 2).*

**Response:** We agreed with this suggestion and revised the text (p. 3–4, lines 72–74).

Line 79. *There is increasing evidence that periodontal disease leads to tooth loss and*
*oral functional decline, as well as the onset or progression of various systemic diseases*
*(5-9).* We know that periodontitis leads to tooth loss and oral functional decline, thus I
suggest the following adjustment to the wording. *Periodontitis leads to tooth loss and*
*oral functional decline, and there is increasing evidence that it is associated with onset*
*or progression of various systemic diseases (5-9).*

**Response:** In accordance with this comment, we revised the text (p. 4, lines 85–87).

**5. Line 501: Table 1 legend. Include the information that age and standard**
**deviation are in years.**

**Response:** We added years in Table 1.

**6. Line 130-131 Table S1 shows data at the species level, not genus level as**
**described in the text, which is a bit confusing. Please explain or modify.**

**Response:** The description in the text was wrong. We are sorry for any confusion
caused. Ideally, we would have added supplemental tables for data at the genus level, but
because there is a limitation to the number of supplemental tables/figures that can be
posted, we decided to remove the original Table S1 (p. 6, lines 130–131) to avoid any
confusion.

**7. Lines 123-125 These total numbers of reconstructed OTUs seems low compared**

**to DNA- based methods. Are these values consistent with other transcriptomics**
**studies?**

**Response:** We checked the numbers and confirmed that high numbers of reconstructed
OTUs were observed with DNA-based methods in our previous studies (Ikeda *et al.*
*Odontology*. 2020 Apr;108(2):280-291., Komatsu *et al.* *Front Cell Infect Microbiol.*
2020 Dec 11;10:596490), whereas low numbers of reconstructed OTUs have been
observed with RNA-based methods (Shiba *et al.* *Sci Rep*. 2016 Aug 8;6:30997,
Funahashi *et al.* *Prog Orthod*. 2019 Mar 25;20(1):11). Bacterial DNA samples used in
meta 16S and metagenome analyses are generally derived from live and dead cells, which
could potentially increase the richness of a bacterial community.

**8. Line 198-199 *The total number of taxa in H, G, and P sites were 2,058, 2,279, and***
***2,129, respectively; These seems like an extremely high number, please describe***
***briefly how taxa are defined here (at the genus level, species level, subspecies level,***
***or a mix?)***

**Response:** We modified the sentence to explain that the taxa were defined at the species
level (p. 9, line 197). In previous studies, we used the NCBI nr database to thoroughly
identify the taxonomic origins of the mRNAs (Shiba *et al.* *Sci Rep*. 2016 Aug 8;6:30997,
Funahashi *et al.* *Prog Orthod*. 2019 Mar 25;20(1):11, Komatsu *et al.* *Front Cell Infect*
*Microbiol.* 2020 Dec 11;10:596490). Unlike the Human Oral Microbiome Database, the
nr database covers a wide range of microbials from many different environments, which
explains why high numbers of taxa were identified in the present study.

**9. Lines 210 -211. *Among active taxa, the activities of eight taxa (F. fastidiosum,***
***Eubacterium nodatum, F. alocis, Actinomyces sp., Prevotella sp., E. saphenum,***
***Porphyromonas endodontalis, and P. nigrescens) were greater than the activities of***
***red complex bacteria in P site. Comparing this text to figure 3AB, only***
***F.fastidiosum,Eubacterium nodatum,F.alocis, and Prevotella sp have a higher***
***mRNA/16srRNA ratio than two of the red species complex. This statement should***
***be modified.***

**Response:** We apologize for this error. We have corrected the statement (p. 9, line 209).

**10. Line 225-227 *The significant active taxa (26/36, 45/45, and 20/26 taxa in H, G,***
***and P sites, respectively) were prevalent in all networks (Figure 4A, Table 2, and 227***
***Tables S6–S8). I am not sure I understand this statement correctly. The input data***
***for correlation analysis was the ratio data, so wouldn't you expect the highly active***

**taxa to be present in the network? And they were prevalent in their own network,**
**but not ALL networks? For example, there are 45 taxa in the G network, but they**
**are not all present in the H or P networks? This will require additional clarification**
**in the text.**

**Response:** We agree that this description was misleading. We have revised the statement
to clarify the meaning (p. 10, lines 223–224).

**11. Line 240. *Approximately half of adults over 30 years of age exhibit periodontal***
***disease in North America.* While this is probably true, the study was done in**
**Japanese adults and in fact periodontal disease is common world-wide. Therefore,**
**this is not the best opening statement for the discussion, in my opinion, maybe**
**something about occurrence world-wide is better.**

**Response:** We agree with this comment and have revised the text to make it more
general (p. 10, lines 236–238).

September 20, 2021

Dr. Yasuo Takeuchi
Tokyo Medical and Dental University
Department of Periodontology, Graduate School of Medical and Dental Sciences
Tokyo
Japan

Re: mSystems00886-21R1 (Discrimination of Bacterial Community Structure among Healthy, Gingivitis, and Periodontitis Statuses through Integrated Metatranscriptomic and Network Analyses)

Dear Dr. Yasuo Takeuchi:

I am satisfied that the authors have addressed all remaining reviewer concerns, and I am now happy to recommend final acceptance for this manuscript.

Your manuscript has been accepted, and I am forwarding it to the ASM Journals Department for publication. For your reference, ASM Journals' address is given below. Before it can be scheduled for publication, your manuscript will be checked by the mSystems senior production editor, Ellie Ghatineh, to make sure that all elements meet the technical requirements for publication. She will contact you if anything needs to be revised before copyediting and production can begin. Otherwise, you will be notified when your proofs are ready to be viewed.

As an open-access publication, mSystems receives no financial support from paid subscriptions and depends on authors' prompt payment of publication fees as soon as their articles are accepted. =

Publication Fees:

- Minimum resolution of 1280 x 720
- .mov or .mp4. video format
- Provide video in the highest quality possible, but do not exceed 1080p

- Provide a still/profile picture that is 640 (w) x 720 (h) max
- Provide the script that was used

We recognize that the video files can become quite large, and so to avoid quality loss ASM suggests sending the video file via <https://www.wetransfer.com/>. When you have a final version of the video and the still ready to share, please send it to Ellie Ghatineh at eghatineh@asmusa.org.

Sincerely,

Holly Bik
Editor, mSystems

Journals Department
Result S1: Accept
Table. S2: Accept
Fig. S4: Accept
Table. S3: Accept
Table. S4: Accept
Fig. S2: Accept
Fig. S3: Accept
Fig. S5: Accept
Fig. S1: Accept
Table. S1: Accept